# Risk factor control and cardiovascular events in patients with type 2 diabetes mellitus

**Do Kyeong Song, Young Sun Hong, Yeon-Ah Sung, Hyejin Lee** *

Department of Internal Medicine, Ewha Womans University School of Medicine, Seoul, Korea

* hyejinlee@ewha.ac.kr

**Data Availability Statement:** All relevant data are within the manuscript and its Supporting Information files.

**Funding:** This work was supported by a grant (D.K. S., 2021F-9) from the Korean Diabetes Association.

## Abstract

### Background

Since patients with type 2 diabetes mellitus (T2DM) have an increased risk of cardiovascular events, interventions addressing risk factors reduce the incidence of cardiovascular disease (CVD) events. This study aimed to evaluate the difference in the incidence of CVD events according to risk factor control in patients with diabetes with and without cardio-renal disease.

### Methods

We analyzed 113,909 patients with diabetes and 290,339 without diabetes using data released by the National Health Insurance Service (NHIS).

### Results

Among patients with diabetes with four or five poorly controlled risk factors, hazard ratio for CVD events was 1.19 (95% confidence interval [CI], 1.06–1.34) in patients with cardio-renal disease and 2.31 (95% CI, 1.95–2.74) in patients without cardio-renal disease compared to patients with diabetes without risk factors. In subjects with diabetes and cardio-renal disease, patients with four or five poorly controlled risk factors had a higher risk of CVD mortality compared to subjects without risk factors (hazard ratio, 1.64; 95% CI, 1.18–2.30).

### Conclusion

Controlling cardiovascular risk factors reduced the incidence of CVD events in patients with diabetes, especially those without cardio-renal disease. The degree of risk control was strongly associated with CVD mortality in patients with diabetes with baseline cardio-renal disease.

## Introduction

Type 2 diabetes mellitus (T2DM) is a complex disorder associated with an increased risk of macro- or microvascular complications and decreased life expectancy [1,2]. Although a

The funders had no role in study design, data collection and analysis, decision to publish, or preparation of the manuscript.

**Competing interests:** The authors have declared that no competing interests exist.

continuous reduction in cardiovascular complications and mortality has been observed among patients with diabetes [3], they still have higher all-cause and cardiovascular mortality rates than patients without diabetes. Cardiovascular mortality remains the leading cause of death among patients with diabetes in Korea [4].

Multiple modifiable risk factors, such as smoking, dysglycemia, high blood pressure, and dyslipidemia, increase the risk of cardiovascular disease (CVD). Fortunately, multiple cardiovascular risk factor interventions, including drug combinations and behavior modifications, reduced the incidence of cardiovascular events and mortality among patients with diabetes with albuminuria in the Steno-2 Study [5–7]. Among three multiethnic U.S. prospective studies including subjects with diabetes without known CVD at baseline, a lower CVD risk was observed when multiple risk factors were well controlled [8]. In a retrospective cohort study using data from an English and Scottish population, even optimally managed patients with diabetes had a higher risk of cardiovascular events compared to subjects without diabetes, and the association between risk factor levels and CVD outcomes was much stronger in patients with diabetes without cardio-renal complications compared to those with such complications [9].

Despite the established benefit of reducing cardiovascular complications by controlling multiple risk factors in patients with diabetes, quantifying the effects of controlling multiple risk factors is difficult in this population according to baseline CVD risk. Thus, it remains unclear whether T2DM directly affects CVD risk in optimally controlled subjects and whether baseline CVD risk influences its association with risk factor control among patients with diabetes. Cumulative evidence suggests that clinical features, complications, and cause-specific mortality differed according to T2DM patients' ethnicity, possibly due to different socioeconomic factors and diabetes management guidelines used in studied countries [1,10].

This study aimed to compare the incidence of CVD events between patients with diabetes without risk factors and those without diabetes and evaluate the association between cardiovascular risk factor control and the incidence of CVD events or mortality based on baseline CVD risk defined by the presence of cardio-renal disease in Korean patients with diabetes.

## Materials and methods

### Data source

We used data from the National Health Insurance Service-National Health Screening Cohort (NHIS-HEALS) between January 1, 2009, and December 31, 2019. Data were assessed for research purposes from February 3, 2022, to March 2, 2022. The NHIS-HEALS is a population-based cohort of subjects who participated in a health screening program based on nationwide health insurance data in Korea. Detailed information on the NHIS can be found elsewhere [11].

We did not obtain informed consent from the participants because the study data had already been collected. Patient records were anonymized before release. The Institutional Review Board (IRB) of Ewha Womans University Mokdong Hospital (approval no. 2021-02-031) approved our study.

### Study population

S1 Fig shows participant selection and entry in this study. We analyzed 113,909 patients with diabetes and 290,339 subjects without diabetes aged >40 years who underwent regular health check-ups using data released by NHIS between January 1, 2009, and December 31, 2019. Participants were observed to the earliest occurrence of a CVD event, death, or last follow-up on December 31, 2019.

T2DM was defined per the E11 code of the 10$^{th}$ edition of the International Classification of Diseases (ICD-10). Subjects without diabetes were defined as those without E10–14 or a prescription for oral glucose-lowering medications or insulin. Baseline cardio-renal disease was defined as prior CVD or estimated glomerular filtration rate (eGFR) < 45 mL/min/1.73 m$^2$ at cohort entry.

## Outcome variables and covariates

Well-controlled risk factors were defined as (1) no smoking, (2) total cholesterol ≤154 mg/dL, (3) triglycerides ≤150 mg/dL, (4) fasting glucose <130 mg/dL, and (5) systolic blood pressure <140 mm Hg or <130 mm Hg with baseline cardio-renal disease or retinopathy [12,13].

The outcomes were time to CVD death and time to CVD occurrence, defined as the first admission due to CVD. Cardiovascular events were defined as composite outcomes of coronary heart disease and stroke, as well as hospitalization for heart failure. CVD was identified based on ICD-10 codes (ICD codes I20–I25 for coronary heart disease, ICD codes I60–I64 for stroke, and ICD code I50 for hospitalization for heart failure).

Drug prescriptions at index date were defined as prescriptions for >30 days. Sociodemographic (age, gender, and smoking history), physical examination (body mass index and blood pressure), laboratory test (fasting glucose, total cholesterol, and triglycerides), and treatment (use of hypoglycemic agents, antihypertensives, and lipid-lowering therapies) data were identified at the index date. The time of health screening after January 1, 2009, was used as the index date, and the follow-up period was defined as the length of time from the index date to the occurrence of CVD, death, or the end date of the study, December 31, 2019.

## Statistical analysis

Baseline characteristics were presented as the mean ± standard deviations for continuous variables. Categorical variables were presented as frequency and proportion. We stratified patients with diabetes by the number of poorly controlled risk factors and baseline presence of cardio-renal disease. We used Cox proportional hazards models to evaluate the risk of CVD events or mortality after adjustments for age, gender, follow-up, history of CVD, and prescriptions for hypoglycemic agents, antihypertensives, and lipid-lowering therapies. Then, we calculated the adjusted hazard ratios (HRs) and 95% confidence intervals. CVD history was not adjusted in the analysis stratified by the baseline presence of cardio-renal disease. $P$ values <0.05 were considered statistically significant. All statistical analyses were performed using SAS (version 9.4, SAS Institute, Cary, NC).

## Results

Baseline characteristics are shown in Tables 1 and 2. Patients with diabetes were older and had a higher body mass index, triglycerides, blood glucose, and blood pressure than those without diabetes. The proportions of current smokers and patients with elevated total cholesterol levels were lower among those with diabetes than in those without diabetes. Patients with diabetes received more antihypertensive and lipid-lowering medications than subjects without diabetes. Patients with diabetes without any risk factors constituted only 7% of the population (n = 8,280). More patients with diabetes had previous CVD and renal impairment compared to those without diabetes; specifically, 52,234 (46%) patients had a history of previous CVD, and 4,168 (4%) patients had eGFR <45 mL/min/1.73 m$^2$ (Table 1).

Among patients with diabetes, 47% (n = 53,858) of them had baseline cardio-renal disease. Patients with diabetes with cardio-renal disease were older and received more antihypertensive and lipid-lowering medications compared to those without cardio-renal disease. Patients with

**Table 1. Baseline clinical characteristics of study participants.**

|  | Patients with diabetes | Subjects without diabetes |
|---|---|---|
| n | 113,909 | 290,339 |
| Age (yrs) | 63.1 ± 9.2 | 57.9 ± 8.5 |
| Women, n (%) | 53,785 (47.2) | 133,511 (46.0) |
| BMI (kg/m$^2$) | 24.4 ± 3.1 | 23.8 ± 2.9 |
| CVD, n (%) | 52,234 (45.9) | 53,251 (18.3) |
| Renal impairment, n (%) | 4,168 (3.7) | 7,229 (2.5) |
| Current smoking, n (%) | 17,430 (15.3) | 53,171 (18.3) |
| Total cholesterol (mg/dL) | 194.1 ± 40.0 | 202.2 ± 36.8 |
| Triglycerides (mg/dL) | 146.8 ± 94.6 | 136.4 ± 89.0 |
| Fasting glucose (mg/dL) | 112.2 ± 34.6 | 96.4 ± 17.3 |
| Systolic blood pressure (mmHg) | 127.5 ± 15.4 | 124.6 ± 15.3 |
| Diastolic blood pressure (mmHg) | 77.9 ± 9.9 | 77.5 ± 10.1 |
| Raised total cholesterol, n (%) | 96,116 (84.4) | 265,956 (91.6) |
| Raised triglycerides, n (%) | 40,003 (35.1) | 87,428 (30.1) |
| Raised fasting glucose, n (%) | 21,729 (19.1) | 8,242 (2.8) |
| Raised blood pressure, n (%) | 17,912 (15.7) | 32,614 (11.2) |
| Hypoglycemic agents, any (%) | 36,465 (32.0) | 0 (0.0) |
| Antihypertensive agents, any (%) | 66,088 (58.0) | 75,068 (25.9) |
| Lipid-lowering therapy, any (%) | 43,082 (37.8) | 35,814 (12.3) |

Data are presented as the means ± standard deviations or the frequencies and proportions.

BMI, body mass index; CVD, cardiovascular disease.

**Table 2. Baseline clinical characteristics of patients with diabetes.**

|  | Patients with diabetes with cardio-renal disease | Patients with diabetes without cardio-renal disease |
|---|---|---|
| n | 53,858 | 60,051 |
| Age (yrs) | 65.4 ± 9.2 | 61.0 ± 8.7 |
| Women, n (%) | 26,348 (48.9) | 27,437 (45.7) |
| BMI (kg/m$^2$) | 24.6 ± 3.2 | 24.3 ± 3.0 |
| Current smoking, n (%) | 6,953 (12.9) | 10,477 (17.5) |
| Total cholesterol (mg/dL) | 190.0 ± 40.6 | 197.8 ± 39.1 |
| Triglycerides (mg/dL) | 145.7 ± 91.5 | 147.8 ± 97.2 |
| Fasting glucose (mg/dL) | 110.6 ± 32.5 | 113.6 ± 36.3 |
| Systolic blood pressure (mmHg) | 128.2 ± 15.5 | 126.9 ± 15.2 |
| Diastolic blood pressure (mmHg) | 77.8 ± 9.9 | 78.0 ± 9.8 |
| Raised total cholesterol, n (%) | 43,640 (81.0) | 52,476 (87.4) |
| Raised triglycerides, n (%) | 18,665 (34.7) | 21,338 (35.5) |
| Raised fasting glucose, n (%) | 9,312 (17.3) | 12,417 (20.7) |
| Raised blood pressure, n (%) | 9,648 (17.9) | 8,264 (13.8) |
| Hypoglycemic agents, any (%) | 17,623 (32.7) | 18,842 (31.4) |
| Antihypertensive agents, any (%) | 39,458 (73.3) | 26,630 (44.4) |
| Lipid-lowering therapy, any (%) | 25,811 (47.9) | 17,271 (28.8) |

Data are presented as the means ± the standard deviations or the frequencies and proportions.

BMI, body mass index.

diabetes with cardio-renal disease were less likely to have elevated total cholesterol and triglyceride levels and more likely to have elevated blood pressure compared to those without cardio-renal disease. Additionally, patients with diabetes with cardio-renal disease were less likely to be current smokers than those without cardio-renal disease (Table 2).

Patients with diabetes had higher CVD event or mortality rates compared to those without diabetes. CVD events occurred in 14,184 (13%) patients with diabetes, with a mean follow-up of five years, and in 18,476 (6%) subjects without diabetes. The rate of coronary heart disease incidence was 7% (n = 7,583), that of stroke incidence was 6% (n = 6,298), and that of hospitalization for heart failure was 1% (n = 1,401) among patients with diabetes. Death due to CVD occurred in 1,766 (1.6%) people with diabetes and 1,896 (0.7%) people without diabetes. Among patients with diabetes, 772 (0.7%) subjects died due to coronary heart disease, 763 (0.7%) subjects died due to stroke, and 231 (0.2%) subjects died due to heart failure (S1 Table).

The HR for all CVD events, coronary heart disease, stroke, and heart failure was 1.13 (P < 0.001), 1.22 (P < 0.001), 0.94 (P = 0.188), and 1.54 (P < 0.001), respectively, among patients with diabetes without risk factors compared to people without diabetes. Patients with diabetes who had four or five poorly controlled risk factors had the HR for all CVD events of 1.19 (P = 0.004) if they had baseline cardio-renal disease and 2.31 (P < 0.001) if they had no baseline cardio-renal disease compared to patients with diabetes without risk factors. Among patients with diabetes and cardio-renal disease, the HR for coronary heart disease, stroke, and hospitalization for heart failure was 1.01 (P = 0.919), 1.57 (P < 0.001), and 1.25 (P = 0.217), respectively, for those with four or five poorly controlled risk factors compared to those without risk factors. In patients with diabetes without cardio-renal disease, the HR for coronary heart disease, stroke, and hospitalization for heart failure was 2.16 (P < 0.001), 2.79 (P < 0.001), and 1.34 (P = 0.320), respectively, for those with four or five poorly controlled risk factors compared to those without risk factors (Table 3 and S2–S4 Tables).

Compared to patients without diabetes, people with diabetes without risk factors had the HR for total cardiovascular mortality, coronary heart disease mortality, stroke mortality, and heart failure mortality of 1.07 (P = 0.444), 1.05 (P = 0.725), 1.04 (P = 0.772), and 1.27 (P = 0.313), respectively. Patients with diabetes and cardio-renal disease who had four or five poorly controlled risk factors had higher CVD mortality compared to those without risk factors (HR, 1.64; P = 0.004). Among patients with diabetes and cardio-renal disease, the HR for coronary heart disease mortality, stroke mortality, and heart failure mortality was 1.62 (P = 0.050), 1.90 (P = 0.016), and 0.99 (P = 0.986), respectively, in individuals with four or five poorly controlled risk factors compared to those without risk factors. However, risk factor control was not associated with CVD mortality in patients with diabetes without cardio-renal disease (Table 4 and S5–S7 Tables).

## Discussion

In our study, patients with diabetes without risk factors had a higher risk of cardiovascular events compared to people without diabetes. Patients with diabetes without any risk factors constituted only 7% of patients with diabetes. Controlling cardiovascular risk factors significantly reduced the incidence of CVD events among patients with diabetes, especially in those without cardio-renal disease. Controlling cardiovascular risk factors reduced CVD mortality in patients with diabetes and baseline cardio-renal disease.

Patients with diabetes and controlled risk factors had a higher risk (13%) of cardiovascular events compared to subjects without diabetes. In an English and Scottish study, patients with diabetes without risk factors (smoking, elevated total cholesterol level, elevated triglycerides, elevated HbA1c level, and elevated blood pressure) had a 21% higher risk of all CVD events

**Table 3. The relative risk of cardiovascular events in participants according to the degree of risk factor control.**

| | | Uncontrolled risk factors, N | Total cases | Events | Person-years | Incidence rate per 1000 person-years (95% CI) | HR | 95% CI | P-value |
|---|---|---|---|---|---|---|---|---|---|
| Total participants | Subjects without diabetes | | 290,339 | 18,476 | 2,622,168 | 7.0 (6.9–7.1) | | | |
| | Patients with diabetes | 0 | 8,280 | 1,156 | 63,454 | 18.2 (17.2–19.3) | 1.13 | 1.06–1.20 | <0.001 |
| | | 1 | 45,253 | 4,780 | 377,932 | 12.6 (12.3–13.0) | 1.04 | 1.00–1.07 | 0.034 |
| | | 2 | 38,348 | 4,871 | 316,143 | 15.4 (15.0–15.8) | 1.18 | 1.14–1.22 | <0.001 |
| | | 3 | 17,264 | 2,597 | 139,275 | 18.6 (17.9–19.4) | 1.43 | 1.36–1.49 | <0.001 |
| | | ≥4 | 4,764 | 780 | 37,838 | 20.6 (19.2–22.1) | 1.67 | 1.54–1.80 | <0.001 |
| Patients with diabetes | | 0 | 8,280 | 1,156 | 63,454 | 18.2 (17.2–19.3) | | | |
| | | 1 | 45,253 | 4,780 | 377,932 | 12.6 (12.3–13.0) | 0.91 | 0.85–0.97 | 0.003 |
| | | 2 | 38,348 | 4,871 | 316,143 | 15.4 (15.0–15.8) | 1.03 | 0.97–1.10 | 0.362 |
| | | 3 | 17,264 | 2,597 | 139,275 | 18.6 (17.9–19.4) | 1.25 | 1.17–1.34 | <0.001 |
| | | ≥4 | 4,764 | 780 | 37,838 | 20.6 (19.2–22.1) | 1.46 | 1.33–1.60 | <0.001 |
| Patients with diabetes with cardio-renal disease | | 0 | 4,859 | 948 | 34,431 | 27.5 (25.8–29.3) | | | |
| | | 1 | 21,305 | 3,286 | 167,401 | 19.6 (19.0–20.3) | 0.87 | 0.81–0.94 | <0.001 |
| | | 2 | 18,147 | 3,161 | 141,258 | 22.4 (21.6–23.2) | 0.94 | 0.87–1.01 | 0.089 |
| | | 3 | 7,698 | 1,583 | 57,950 | 27.3 (26.0–28.7) | 1.12 | 1.03–1.21 | 0.009 |
| | | ≥4 | 1,849 | 400 | 13,720 | 29.2 (26.3–32.0) | 1.19 | 1.06–1.34 | 0.004 |
| Patients with diabetes without cardio-renal disease | | 0 | 3,421 | 208 | 29,023 | 7.2 (6.2–8.1) | | | |
| | | 1 | 23,948 | 1,494 | 210,531 | 7.1 (6.7–7.5) | 1.14 | 0.98–1.32 | 0.081 |
| | | 2 | 20,201 | 1,710 | 174,885 | 9.8 (9.3–10.2) | 1.44 | 1.24–1.66 | <0.001 |
| | | 3 | 9,566 | 1,014 | 81,325 | 12.5 (11.7–13.2) | 1.78 | 1.53–2.07 | <0.001 |
| | | ≥4 | 2,915 | 380 | 24,118 | 15.8 (14.2–17.3) | 2.31 | 1.95–2.74 | <0.001 |

Hazard ratios were adjusted for age, gender, follow-up, history of cardiovascular disease, and prescriptions for hypoglycemic, antihypertensive, and lipid-lowering therapy.

HR, hazard ratio; CI, confidence interval.

compared to those without diabetes [9], which is similar to our data. In contrast, the risk of myocardial infarction or stroke was not increased in patients with diabetes without risk factors (elevated HbA1c level, elevated low-density lipoprotein cholesterol level, albuminuria, smoking, and elevated blood pressure); however, the risk of hospitalization for heart failure was higher among these subjects compared to those without diabetes according to the Swedish National Diabetes Register. A Swedish study excluded patients with a previous stroke, acute myocardial infarction, or amputation, those who had undergone dialysis or renal transplantation, and those with a body mass index <18.5 kg/m$^2$ [14]. Differences in study design and inclusion or exclusion criteria might contribute to the differences in CVD outcomes noted in these studies.

Intensified multiple risk factor control reduced the risk of stroke and hospitalization for heart failure in 160 Danish patients of European descent with T2DM and albuminuria during 21 years of follow-up in the Steno-2 Study [15,16]. In the ACCORD BP trial, intensive blood pressure or glycemia treatment improved CVD outcomes compared with standard treatment in patients with diabetes across the U.S. and Canada [17]. Consistent with the above-mentioned results, we found that cardiovascular risk factor control reduced the incidence of CVD events in patients with diabetes. In contrast, another study found no difference in CVD incidence according to the degree of risk factor control in patients with diabetes. In Japan, the rate of acute coronary events did not differ between patients receiving intensified multifactorial

**Table 4. The relative risk of cardiovascular mortality in participants according to the degree of risk factor control.**

| | | Uncontrolled risk factors, N | Total cases | Events | Person-years | Incidence rate per 1000 person-years (95% CI) | HR | 95% CI | P-value |
|---|---|---|---|---|---|---|---|---|---|
| Total participants | Subjects without diabetes | | 290,339 | 1,896 | 2,699,573 | 0.7 (0.6–0.7) | | | |
| | Patients with diabetes | 0 | 8,280 | 149 | 68,385 | 2.2 (1.8–2.5) | 1.07 | 0.90–1.27 | 0.444 |
| | | 1 | 45,253 | 602 | 397,662 | 1.5 (1.4–1.6) | 1.01 | 0.91–1.11 | 0.868 |
| | | 2 | 38,348 | 606 | 336,502 | 1.8 (1.7–1.9) | 1.14 | 1.03–1.26 | 0.013 |
| | | 3 | 17,264 | 329 | 150,339 | 2.2 (2.0–2.4) | 1.51 | 1.33–1.72 | <0.001 |
| | | ≥4 | 4,764 | 80 | 41,258 | 1.9 (1.5–2.4) | 1.70 | 1.34–2.14 | <0.001 |
| Patients with diabetes | | 0 | 8,280 | 149 | 68,385 | 2.2 (1.8–2.5) | | | |
| | | 1 | 45,253 | 602 | 397,662 | 1.5 (1.4–1.6) | 0.92 | 0.76–1.10 | 0.344 |
| | | 2 | 38,348 | 606 | 336,502 | 1.8 (1.7–1.9) | 1.03 | 0.86–1.24 | 0.720 |
| | | 3 | 17,264 | 329 | 150,339 | 2.2 (2.0–2.4) | 1.38 | 1.13–1.67 | 0.001 |
| | | ≥4 | 4,764 | 80 | 41,258 | 1.9 (1.5–2.4) | 1.55 | 1.18–2.04 | 0.002 |
| Patients with diabetes with cardio-renal disease | | 0 | 4,859 | 112 | 38,650 | 2.9 (2.4–3.4) | | | |
| | | 1 | 21,305 | 452 | 181,006 | 2.5 (2.3–2.7) | 1.03 | 0.83–1.27 | 0.798 |
| | | 2 | 18,147 | 411 | 154,638 | 2.7 (2.4–2.9) | 1.05 | 0.85–1.29 | 0.664 |
| | | 3 | 7,698 | 214 | 64,729 | 3.3 (2.9–3.7) | 1.39 | 1.11–1.75 | 0.005 |
| | | ≥4 | 1,849 | 51 | 15,464 | 3.3 (2.4–4.2) | 1.64 | 1.18–2.30 | 0.004 |
| Patients with diabetes without cardio-renal disease | | 0 | 3,421 | 37 | 29,735 | 1.2 (0.8–1.6) | | | |
| | | 1 | 23,948 | 150 | 216,655 | 0.7 (0.6–0.8) | 0.67 | 0.46–0.96 | 0.028 |
| | | 2 | 20,201 | 195 | 181,864 | 1.1 (0.9–1.2) | 0.96 | 0.68–1.37 | 0.820 |
| | | 3 | 9,566 | 115 | 85,610 | 1.3 (1.1–1.6) | 1.26 | 0.87–1.82 | 0.229 |
| | | ≥4 | 2,915 | 29 | 25,795 | 1.1 (0.7–1.5) | 1.30 | 0.79–2.12 | 0.303 |

Hazard ratios were adjusted for age, gender, follow-up, history of cardiovascular disease, and prescriptions for hypoglycemic, antihypertensive, and lipid-lowering therapy.

HR, hazard ratio; CI, confidence interval.

interventions and standard care; however, cerebrovascular events were more seldom in the intensive therapy group compared to the conventional therapy group among 2,542 people aged 45–69 years with diabetes who had hypertension, dyslipidemia, or both. Because glucose, blood pressure, and lipids were well controlled even in the conventional therapy group, the benefit of multiple risk factor control might have been lower than anticipated in the intensive therapy group in this population [18].

Controlling cardiovascular risk factors reduced the incidence of CVD events in patients with diabetes with and without cardio-renal disease. However, the association between risk factor control and the incidence of CVD events was stronger in patients with diabetes without cardio-renal disease compared to those with cardio-renal disease in our study. Although the responsible mechanism is unclear, patients with diabetes without cardio-renal disease were younger and less likely to use antihypertensive and lipid-lowering medications compared to patients with diabetes and cardio-renal disease. Consistent with our results, a previous study using the NHIS database found a positive relationship between blood pressure and CVD risk in relatively low–risk people with diabetes without previous CVD, with a greater association for younger age groups [19]. Likewise, multiple risk factor control incrementally reduced the incidence of total CVD among Chinese subjects with diabetes without previous CVD. Controlled risk factors were the most beneficial in patients with diabetes mellitus for <1 year [20]. The UK Prospective Diabetes Study (UKPDS) found that the risk of developing CVD was

additively associated with hyperglycemia and hypertension among patients newly diagnosed with diabetes without severe vascular diseases, such as myocardial infarction or stroke, within the previous year [21]. Therefore, risk factors should be timely controlled to prevent CVD development in patients with diabetes, even without cardio-renal disease.

Meanwhile, cardiovascular risk factor control reduced CVD mortality in patients with diabetes with cardio-renal disease but not in those without cardio-renal disease. Similar to the results of our study, the number of poorly controlled risk factors (smoking, non-high-density lipoprotein cholesterol, triglycerides, systolic blood pressure, diastolic blood pressure, and HbA1c) was strongly associated with cardiovascular morbidity and mortality in patients with diabetes and angiographically documented stable coronary heart disease in the BARI 2D trial [22]. Therefore, more intensive risk factor control can reduce the risk of fatal CVD in patients with diabetes with cardio-renal disease. Furthermore, a previous study indicated that regulation of low-density lipoprotein cholesterol, a modifiable risk factor for CVD, was suboptimal in Korean patients with diabetes [23]. Patients with diabetes without any risk factors constituted only 7% of the population in our study. Furthermore, only 10% of Chinese patients with diabetes had no risk factors (HbA1c, blood pressure, and low-density lipoprotein cholesterol) [20], and only 6% of an English and Scottish cohort with diabetes had well-controlled risk factors [9]. Therefore, risk factors must be well managed in patients with diabetes.

This is the first study to evaluate the association between cardiovascular risk factor control and the incidence of CVD events or mortality according to the prevalence of cardio-renal disease in Korean patients with diabetes. The results of our study can be generalized to Korean patients with diabetes. We adjusted multiple covariates influencing CVD development, such as age, gender, follow-up, history of CVD, and prescriptions for hypoglycemic, antihypertensive, and lipid-lowering therapy.

There were several limitations in our study. Because this was a retrospective study, clarifying the causative relationship between risk factor control and CVD outcomes among patients with diabetes according to the prevalence of cardio-renal disease was difficult. We did not estimate the presence of albuminuria, which is required for the diagnosis of diabetic nephropathy that can be related to CVD development, due to limited data. Because the diagnosis of diabetes was based on study entry, the subjects who were diagnosed with T2DM during the follow-up might have been misclassified as subjects without diabetes. Potential residual confounding factors, such as lifestyle (diet and exercise) or socioeconomic factors, might have influenced the association between risk factor control and cardiovascular outcomes in this study population. We did not adjust for diabetes duration or variables reflecting the severity of diabetes, such as HbA1c or fasting glucose. Although the baseline period was long, we did not include the index year in the multivariable adjusted model. Furthermore, caution might be needed in interpreting the results because we applied the cut-off value for CVD risk factors by our own reference for the range of control. Total cholesterol and systolic blood pressure were included in the study to simply identify the risk factors to control. Further studies defining well-controlled risk factors based on low-density lipoprotein cholesterol levels or diastolic blood pressure are needed. We did not consider risk factor changes during the follow-up and treatment with antiplatelet drugs that could influence the prevalence of CVD events. Further studies are warranted to evaluate the effect of changes in risk factors on CVD outcomes during diabetes progression and considering treatment with antiplatelet drugs. The occurrence of CVD was evaluated using the ICD-10 codes, which might differ from the actual number. Finally, patients with diabetes in our study might not have used newer hypoglycemic agents protective against CVD events, such as sodium-glucose cotransporter 2 inhibitors or glucagon-like peptide-1 receptor agonists [24,25]. CVD outcomes were not considered according to the type of hypoglycemic agents used.

In conclusion, cardiovascular risk factor control significantly reduced the incidence of CVD events among patients with diabetes, especially in those without cardio-renal disease. There was a strong association between the degree of risk control and CVD mortality in patients with diabetes and cardio-renal disease. Thus, early and more intensive cardiovascular risk factor interventions, including educating and promoting the management of modifiable risk factors, help reduce the incidence of cardiovascular events and mortality in Korean patients with diabetes.

## Supporting information

**S1 Fig. Study populations from the National Health Insurance Service database.** CVD, cardiovascular disease.
(TIF)

**S1 Table. Cardiovascular outcomes of study participants.** Data are presented as frequencies and proportions. CVD, cardiovascular disease.
(DOCX)

**S2 Table. The relative risk of coronary events in participants according to the degree of risk factor.** Hazard ratios were adjusted for age, gender, follow-up, history of cardiovascular disease, and prescriptions for hypoglycemic, antihypertensive, and lipid-lowering therapy. HR, hazard ratio; CI, confidence interval.
(DOCX)

**S3 Table. The relative risk of cerebrovascular events in participants according to the degree of risk factor.** Hazard ratios were adjusted for age, gender, follow-up, history of cardiovascular disease, and prescriptions for hypoglycemic, antihypertensive, and lipid-lowering therapy. HR, hazard ratio; CI, confidence interval.
(DOCX)

**S4 Table. The relative risk of heart failure hospitalization in participants according to the degree of risk factor.** Hazard ratios were adjusted for age, gender, follow-up, history of cardiovascular disease, and prescriptions for hypoglycemic, antihypertensive, and lipid-lowering therapy. HR, hazard ratio; CI, confidence interval.
(DOCX)

**S5 Table. The relative risk of coronary heart disease mortality in participants according to the degree of risk factor.** Hazard ratios were adjusted for age, gender, follow-up, history of cardiovascular disease, and prescriptions for hypoglycemic, antihypertensive, and lipid-lowering therapy. HR, hazard ratio; CI, confidence interval.
(DOCX)

**S6 Table. The relative risk of stroke mortality in participants according to the degree of risk factor.** Hazard ratios were adjusted for age, gender, follow-up, history of cardiovascular disease, and prescriptions for hypoglycemic, antihypertensive, and lipid-lowering therapy. HR, hazard ratio; CI, confidence interval.
(DOCX)

**S7 Table. The relative risk of heart failure mortality in participants according to the degree of risk factor.** Hazard ratios were adjusted for age, gender, follow-up, history of cardiovascular disease, and prescriptions for hypoglycemic, antihypertensive, and lipid-lowering therapy. HR, hazard ratio; CI, confidence interval.
(DOCX)

## Author Contributions

**Conceptualization:** Yeon-Ah Sung, Hyejin Lee.

**Formal analysis:** Do Kyeong Song.

**Funding acquisition:** Do Kyeong Song, Yeon-Ah Sung, Hyejin Lee.

**Investigation:** Do Kyeong Song, Young Sun Hong, Yeon-Ah Sung, Hyejin Lee.

**Methodology:** Do Kyeong Song, Young Sun Hong, Yeon-Ah Sung, Hyejin Lee.

**Project administration:** Do Kyeong Song, Hyejin Lee.

**Supervision:** Young Sun Hong, Yeon-Ah Sung, Hyejin Lee.

**Validation:** Do Kyeong Song.

**Visualization:** Do Kyeong Song.

**Writing – original draft:** Do Kyeong Song.

**Writing – review & editing:** Do Kyeong Song, Young Sun Hong, Yeon-Ah Sung, Hyejin Lee.

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
