## [Decision Letter · Decision Letter 0]

26 Dec 2023

PONE-D-23-37030Risk Factor Control and Cardiovascular Events in Patients with Type 2 Diabetes MellitusPLOS ONE

Dear Dr. Lee,

Thank you for submitting your manuscript to PLOS ONE. After careful consideration, we feel that it has merit but does not fully meet PLOS ONE’s publication criteria as it currently stands. Therefore, we invite you to submit a revised version of the manuscript that addresses the points raised during the review process.

**ACADEMIC EDITOR: **

We look forward to receiving your revised manuscript.

Kind regards,

Hidetaka Hamasaki

Academic Editor

PLOS ONE

 [This work was supported by a grant (D.K.S., 2021F-9) from the Korean Diabetes Association.].  

Reviewers' comments:

Reviewer's Responses to Questions

**Comments to the Author**

1. Is the manuscript technically sound, and do the data support the conclusions?

Reviewer #1: Yes

Reviewer #2: Partly

2. Has the statistical analysis been performed appropriately and rigorously? 

Reviewer #1: Yes

Reviewer #2: Yes

3. Have the authors made all data underlying the findings in their manuscript fully available?

Reviewer #1: Yes

Reviewer #2: Yes

4. Is the manuscript presented in an intelligible fashion and written in standard English?

Reviewer #1: Yes

Reviewer #2: No

5. Review Comments to the Author

Reviewer #1: The present study was aimed at risk factor control and cardiovascular events in patients with type 2 diabetes mellitus. Controlling cardiovascular risk factors reduced CVD events in diabetic patients, especially in low CVD risk group. There was a strong association between the degree of risk control and risk of CVD mortality in diabetic patients with baseline high CVD risk. These findings highlight that early and more intensive cardiovascular risk factor interventions, which help reduce cardiovascular events and mortality in Korean patients with diabetes. The data are reliable and the article is well written. The study is closely integrated with the clinic and has certain innovation.

It would be more comprehensive to add information about pharmacological treatment with antiplatelet drugs. The study was a retrospective study, and future cohort studies should be conducted to support the findings of this study.

Reviewer #2: Authors analyzed national insurance data source and data from one hospital to understand CVD events and risk factors in Korean diabetic patients. The study helps to develop treatment options for diabetes patients based on the risk factors. I have several critiques authors have to address.

1. Is there any reason why authors categorize previous CVD history or eGFR <45 as a high CVD risk in the diabetes population? Is this in the guideline?

2. If the above answer is no, the term of CVD risk is confusing. Authors often use CVD risk, risk factors and CVD events in a sentence. It is really difficult to comprehend authors intentions.

3. Is CDM analysis necessary? Authors admit the limitation of the data such as smaller sample size, tracking mortality and so on. Authors spend significant amounts of CDM analysis in the manuscript. This makes the discussion complicated and wordy. It seems NHIS data is sufficient to lead authors conclusions.

4. Results section is confusing. Readers have to go back and forth among tables. Please rewrite the section or tables to make them in order.

5. Please use standard English. There are many places where I do not comprehend easily what authors try to convey. I would recommend English proofing service to polish manuscript. Below are the examples.

• Authors often use “without poorly controlled risk factors”. Do those simply go with “without risk factors” or “with controlled risk factors”?

• Line 245-246, I do not understand the authors intention to use the sentence here.

• Line 249-252, I do not understand the sentence.

• Line 291-294, I do not understand the sentence.

• Line 316-317, the sentence is difficult to comprehend. Is this “diabetic patients who do not have any risk factors are only 7% in the population”?

• Line 320-322, I do not understand the authors intention for the sentence.

• Line 323-326, the sentence is too long and confusing. Several “and” and “or” are used in one sentence.

• Line 355-357, repetitive.

6. PLOS authors have the option to publish the peer review history of their article (what does this mean?). If published, this will include your full peer review and any attached files.

Reviewer #1: No

Reviewer #2: No

---

## [Author Response · Author response to Decision Letter 0]

27 Jan 2024

→ Thank you for your kind comments. We revised the manuscript to conform to the style requirements of PLOS ONE.

 [This work was supported by a grant (D.K.S., 2021F-9) from the Korean Diabetes Association.]. 

→ We included this amended Role of Funder statement in the cover letter.

→ We included captions for your Supporting Information files at the end of your manuscript.

Reviewers' comments:

Reviewer's Responses to Questions

Comments to the Author

1. Is the manuscript technically sound, and do the data support the conclusions?

Reviewer #1: Yes

Reviewer #2: Partly

2. Has the statistical analysis been performed appropriately and rigorously?

Reviewer #1: Yes

Reviewer #2: Yes

3. Have the authors made all data underlying the findings in their manuscript fully available?

Reviewer #1: Yes

Reviewer #2: Yes

4. Is the manuscript presented in an intelligible fashion and written in standard English?

Reviewer #1: Yes

Reviewer #2: No

5. Review Comments to the Author

Reviewer #1: The present study was aimed at risk factor control and cardiovascular events in patients with type 2 diabetes mellitus. Controlling cardiovascular risk factors reduced CVD events in diabetic patients, especially in low CVD risk group. There was a strong association between the degree of risk control and risk of CVD mortality in diabetic patients with baseline high CVD risk. These findings highlight that early and more intensive cardiovascular risk factor interventions, which help reduce cardiovascular events and mortality in Korean patients with diabetes. The data are reliable and the article is well written. The study is closely integrated with the clinic and has certain innovation.

It would be more comprehensive to add information about pharmacological treatment with antiplatelet drugs. The study was a retrospective study, and future cohort studies should be conducted to support the findings of this study.

→ Thank you for your kind comments. “We did not consider risk factor changes during the follow-up and treatment with antiplatelet drugs that could influence the prevalence of CVD events.” was inserted in the Discussion.

Reviewer #2: Authors analyzed national insurance data source and data from one hospital to understand CVD events and risk factors in Korean diabetic patients. The study helps to develop treatment options for diabetes patients based on the risk factors. I have several critiques authors have to address.

1. Is there any reason why authors categorize previous CVD history or eGFR <45 as a high CVD risk in the diabetes population? Is this in the guideline?

→ Thank you for your kind comments. We changed “High CVD risk” into “cardio-renal disease”. We inserted “Baseline cardio-renal disease was defined as prior CVD or estimated glomerular filtration rate (eGFR) < 45 mL/min/1.73 m2 at cohort entry.” in the Materials and methods.

2. If the above answer is no, the term of CVD risk is confusing. Authors often use CVD risk, risk factors and CVD events in a sentence. It is really difficult to comprehend authors intentions.

→ Thank you for your kind comments. We deleted the term high or low CVD risk. Instead, we stratified the diabetic patients according to the presence of cardio-renal disease.

3. Is CDM analysis necessary? Authors admit the limitation of the data such as smaller sample size, tracking mortality and so on. Authors spend significant amounts of CDM analysis in the manuscript. This makes the discussion complicated and wordy. It seems NHIS data is sufficient to lead authors conclusions.

→ Thank you for your kind comments. We deleted the data of CDM.

4. Results section is confusing. Readers have to go back and forth among tables. Please rewrite the section or tables to make them in order.

→ Thank you for your kind comments. We deleted the data of CDM and rewrited the section to make them in order. 

5. Please use standard English. There are many places where I do not comprehend easily what authors try to convey. I would recommend English proofing service to polish manuscript. Below are the examples.

→ Thank you for your kind comments. We received full English editing.

• Authors often use “without poorly controlled risk factors”. Do those simply go with “without risk factors” or “with controlled risk factors”?

→Thank you for your kind comments. We changed “without poorly controlled risk factors” into “without risk factors”.

• Line 245-246, I do not understand the authors intention to use the sentence here.

→We removed “Diabetic patients with severe disabilities or who were hospitalized might be excluded from the NHIS database.” because CDM data was excluded from the paper.

• Line 249-252, I do not understand the sentence.

→ We removed “It was possible that the risk of CVD did not decrease in patients without diabetes because of other comorbidities or advanced diseases compared to those with diabetes without poorly controlled risk factors per the CDM.” because CDM data was excluded from the paper.

• Line 291-294, I do not understand the sentence.

→We removed the sentence.

• Line 316-317, the sentence is difficult to comprehend. Is this “diabetic patients who do not have any risk factors are only 7% in the population”?

→ We changed the sentence to “Patients with diabetes without any risk factors constituted only 7% of the population.”

• Line 320-322, I do not understand the authors intention for the sentence.

→ We changed “Risk factor control must be promoted to aid in the real-world management of people with diabetes.” to “Therefore, risk factors must be well managed in patients with diabetes.”

• Line 323-326, the sentence is too long and confusing. Several “and” and “or” are used in one sentence.

→ We changed the sentence to “This is the first study to evaluate the association between cardiovascular risk factor control and the incidence of CVD events or mortality according to the prevalence of cardio-renal disease in Korean patients with diabetes.”

• Line 355-357, repetitive. 

→ We removed the sentence “It was difficult to determine the actual duration of diabetes with NHIS or CDM data because the subjects included in the study might not have been diagnosed with diabetes at the beginning of the study.”

6. PLOS authors have the option to publish the peer review history of their article (what does this mean?). If published, this will include your full peer review and any attached files.

Do you want your identity to be public for this peer review? For information about this choice, including consent withdrawal, please see our Privacy Policy.

Reviewer #1: No

Reviewer #2: No

→ We confirm that the figure meets PLOS requirements although the figure was supporting information.

---

## [Decision Letter · Decision Letter 1]

5 Feb 2024

Risk Factor Control and Cardiovascular Events in Patients with Type 2 Diabetes Mellitus

PONE-D-23-37030R1

Dear Dr. Lee,

We’re pleased to inform you that your manuscript has been judged scientifically suitable for publication and will be formally accepted for publication once it meets all outstanding technical requirements.

Kind regards,

Hidetaka Hamasaki

Academic Editor

PLOS ONE

Additional Editor Comments (optional):

Reviewers' comments:

Reviewer's Responses to Questions

**Comments to the Author**

1. If the authors have adequately addressed your comments raised in a previous round of review and you feel that this manuscript is now acceptable for publication, you may indicate that here to bypass the “Comments to the Author” section, enter your conflict of interest statement in the “Confidential to Editor” section, and submit your "Accept" recommendation.

Reviewer #1: All comments have been addressed

Reviewer #2: All comments have been addressed

2. Is the manuscript technically sound, and do the data support the conclusions?

Reviewer #1: Yes

Reviewer #2: Yes

3. Has the statistical analysis been performed appropriately and rigorously? 

Reviewer #1: Yes

Reviewer #2: Yes

4. Have the authors made all data underlying the findings in their manuscript fully available?

Reviewer #1: Yes

Reviewer #2: Yes

5. Is the manuscript presented in an intelligible fashion and written in standard English?

Reviewer #1: Yes

Reviewer #2: Yes

6. Review Comments to the Author

Reviewer #1: The present study was aimed at risk factor control and cardiovascular events in patients with type 2 diabetes mellitus. Controlling cardiovascular risk factors reduced CVD events in diabetic patients, especially in low CVD risk group. There was a strong association between the degree of risk control and risk of CVD mortality in diabetic patients with baseline high CVD risk. These findings highlight that early and more intensive cardiovascular risk factor interventions, which help reduce cardiovascular events and mortality in Korean patients with diabetes. The data are reliable and the article is well written. The study is closely integrated with the clinic and has certain innovation.

Reviewer #2: I congratulate authors. The manuscript has been improved much. Now I think it is good to go for the publication.

7. PLOS authors have the option to publish the peer review history of their article (what does this mean?). If published, this will include your full peer review and any attached files.

Reviewer #1: No

Reviewer #2: No

---

## [Editor Report · Acceptance letter]

21 Feb 2024

PONE-D-23-37030R1 

PLOS ONE

Dear Dr. Lee, 

I'm pleased to inform you that your manuscript has been deemed suitable for publication in PLOS ONE. Congratulations! Your manuscript is now being handed over to our production team.

Kind regards, 

on behalf of

Dr. Hidetaka Hamasaki 

Academic Editor

PLOS ONE